# A Scoping Review of Approaches to Improving Quality of Data Relating to Health Inequalities

**DOI:** 10.3390/ijerph192315874

**Published:** 2022-11-29

**Authors:** Sowmiya Moorthie, Vicki Peacey, Sian Evans, Veronica Phillips, Andres Roman-Urrestarazu, Carol Brayne, Louise Lafortune

**Affiliations:** 1Cambridge Public Health, Interdisciplinary Research Centre, University of Cambridge, Cambridge CB2 OSZ, UK; 2Cambridgeshire County Council, Alconbury, Huntingdon PE28 4YE, UK; 3Local Knowledge Intelligence Service (LKIS) East, Office for Health Improvements and Disparities, UK; 4Medical Library, School of Clinical Medicine, University of Cambridge, Cambridge CB2 0SP, UK

**Keywords:** health inequalities, health disparities, data quality, public health

## Abstract

Identifying and monitoring of health inequalities requires good-quality data. The aim of this work is to systematically review the evidence base on approaches taken within the healthcare context to improve the quality of data for the identification and monitoring of health inequalities and describe the evidence base on the effectiveness of such approaches or recommendations. Peer-reviewed scientific journal publications, as well as grey literature, were included in this review if they described approaches and/or made recommendations to improve data quality relating to the identification and monitoring of health inequalities. A thematic analysis was undertaken of included papers to identify themes, and a narrative synthesis approach was used to summarise findings. Fifty-seven papers were included describing a variety of approaches. These approaches were grouped under four themes: policy and legislation, wider actions that enable implementation of policies, data collection instruments and systems, and methodological approaches. Our findings indicate that a variety of mechanisms can be used to improve the quality of data on health inequalities at different stages (prior to, during, and after data collection). These findings can inform us of actions that can be taken by those working in local health and care services on approaches to improving the quality of data on health inequalities.

## 1. Introduction

Health inequalities are often defined as “differences in health across the population and between different groups” [1]. The study of health inequalities aims to better understand factors that contribute to unfair differences in the status of people’s health to address them and achieve fairer and more inclusive health care. Inequalities in health can arise because of differences in the care that people receive and the opportunities they have to lead healthy lives, including differences in health status (e.g., life expectancy), quality and experience of care, and wider determinants of health [1].

Data analysis to improve understanding of health gaps is an important exercise that contributes to an aspiration for fair and inclusive health. Good data is vital for understanding inequality in health service provision and health outcomes, and necessary for informing and evaluating attempts to improve care or reduce inequality. In the United Kingdom, health inequalities are identified by analysing data across socio-economic factors, geography, and specific characteristics including those protected in law such as sex, ethnicity or disability, and socially excluded groups. However, the quality of data underpinning these analyses can be improved [2,3,4]. Good-quality data are data that are fit for the purpose; therefore, criteria on what constitutes “good” can vary. Dimensions such as completeness, accuracy, relevance, availability, and timeliness of data can be assessed to determine data quality [5].

Several policy reports released in the UK have highlighted the importance of improving the quality of data used for the identification and monitoring of health inequalities [6,7]. In particular, identifying and reducing inequalities linked to ethnicity are a key part of expectations in terms of improving NHS services [8]. Recommendations from these reports include ensuring consistent reporting and analysis of data on ethnicity, health, and health care and documenting and evaluating best practices [6]. The need for better data coverage across all age groups and allowing self-identification, particularly around ethnicity, has also been recommended [9].

Health inequalities have been increasing in England over the past 10 years and the COVID-19 pandemic has starkly highlighted inequalities that exist [10,11]. The pandemic has also demonstrated that collecting data at speed and using healthcare data in flexible and creative ways is possible [12]. This has renewed emphasis on the need for action to address inequalities at both national and system levels. This includes initiatives to improve data and make better use of data to address health inequalities [13,14]. A comprehensive understanding of the evidence base on how data quality can be improved and what has been shown to work is essential to inform the myriad of initiatives in the UK to address health inequalities.

The aim of this work was to identify and review the evidence base on approaches taken within the healthcare context to improve the quality of the data used for the identification and monitoring of health inequalities. The specific objectives were to describe the approaches that have been used or recommended to improve the quality (availability, completeness, accuracy, relevance, and timeliness) of data for identification of health inequalities, to describe the approaches that have been used or recommended to improve the quality of data for monitoring changes in health inequalities, and to describe the evidence base for the effectiveness of such approaches or recommendations.

## 2. Methods

### 2.1. Article Identification and Selection

A systematic literature search was conducted in the databases Medline (via Ovid), Embase (via Ovid), Global Health (via Ebscohost), Cinahl (via Ebschohost), and Web of Science (Core Collection) in September 2021 using search terms relating to data, data quality, and specific terms such as protected characteristics and tailoring them for each database (detailed search terms in Appendix A). Search terms were deliberately broad to maximise the identification of relevant work, as more specific preliminary searches did not identify papers that were already known to be relevant. The searches were not date-limited initially. However, because legislation and guidance in the United Kingdom around recording data on health inequality characteristics has changed considerably in the last 10 years, we subsequently discarded reports published prior to January 2010. Papers that were published after 2010 but reported on data collected before 2010 were included. We undertook citation searching to identify other sources of information not identified in the database searches. In addition, a grey literature search was conducted using the advanced search function in Google. The search terms ‘improving data quality health inequalities’ were used; searches were restricted to .pdf file types. The first five pages of the search were examined for any documents that could be included. All results were limited to the English language.

### 2.2. Inclusion and Exclusion Criteria

Protocols for scoping reviews are not eligible for publication in PROSPERO; however, we have presented our findings as much as possible according to PRISMA guidelines [15] (Appendix A). Peer-reviewed scientific journal publications, as well as grey literature, were included in this review if they described mechanisms to improve data quality relating to the identification and monitoring of health inequalities. Work that focused solely on improving the quality of data on health outcomes was not included, nor was work that simply evaluated data quality rather than presented attempts to improve data quality.

Two reviewers (SM and ARU) independently carried out primary screening of titles and abstracts to identify articles eligible for inclusion based on our eligibility criteria. A third reviewer (LL) reviewed all articles selected for inclusion and those marked as unsure and resolved any disagreements between the reviewers. The decision made by the third reviewer was final for inclusion or exclusion. Two reviewers (LL and SM) screened full-text articles for further assessment of eligibility for inclusion in this review. Discussion was undertaken to resolve any discrepancies. Figure 1 shows the search and selection outcomes for each stage of the review process.

### 2.3. Data Extraction and Synthesis

Data extraction was carried out using an iterative process. One author (SM) read all papers eligible for inclusion, grouped them into broad categories based on the main type of data that was discussed (ethnicity, gender/sexual orientation, social determinants, general, or other), and extracted relevant text from sections of the papers that provided recommendations, methods, or approaches to improving data quality. Papers were categorised as “general” if they did not specify a particular inequality dimension or were across several dimensions. All the papers were subsequently read by at least one other author (VP, SE, or LL) to confirm and supplement the extraction before coding and to ensure quality and consistency. Any discrepancies in data extraction were resolved through discussion. A thematic analysis was then conducted by one reviewer (SM) to identify themes, which were then summarised narratively and validated by another reviewer (LL). There was heterogeneity in the included reports in terms of the subject matter and approaches used. This precluded us from using traditional quality-assurance measures for critiques of the papers.

## 3. Results

The initial database search revealed 21,788 records. Following automatic de-duplication and removal of articles published pre-2010, 7830 articles were identified for primary screening. A total of 110 articles met the eligibility criteria for retrieving full texts after primary screening. A further 27 reports were identified by the grey literature search. Seventy-nine studies were excluded following assessment of full texts. The main reason for exclusion was a lack of discussion on mechanisms to improve data quality. A total of 57 publications were included in the review. Table 1 provides a summary of the characteristics of these reports. Most were peer-reviewed publications (*n* = 49) with the remainder being grey literature (*n* = 8). Many were reporting on data related to the dimension of ethnicity (*n* = 31) or were more general across indicators related to health inequalities (*n =* 15). A smaller number were identified that were focused on dimensions of sexual orientation and gender (*n* = 6), or on specific areas such as infectious diseases, learning disabilities, or cardiovascular care. Most were from the US or UK. None of the studies identified were high on the traditional hierarchy of evidence, and in most cases the approaches that were used for improving data quality had several elements that could not be disaggregated.

### 3.1. Distal Initiatives

The mechanisms and approaches that were upstream of data collection and analysis, but which impacted on these, were grouped under the theme “distal initiatives”. A total of 26 reports stated that policy and legislative imperatives such as mandating data collection led to improvements and consistency in data quality (Table 1). This is through making it a priority and incentivising data collection and leading to the creation of data systems that facilitate such efforts [4,53,58]. Reports also evidenced how data collection had improved since the introduction of mandates and the prioritisation of ethnicity data collection [4,19,31,42,43,45,47,58,65]. In the UK, the Equality Act 2010 and incentivisation under the Quality and Outcomes Framework (QOF) had a significant impact on the completeness of ethnicity data [45,47]. Mathur et al. (2014) [47] describe how the proportion of patients with a valid self-reported ethnicity record changed over time (1995 to 2011) in English hospital data and GP data (via the Clinical Practice Research Datalink, which covered 6% of all GP practices in 2012). The proportion of people with a usable ethnicity recording in Hospital Episode Statistics (HES) inpatient data jumped from 50% to just under 70% in one year between 2000 and 2001. Between 2008 and 2011, the proportion with a usable record also changed from around 20% to around 50% in the HES A&E and outpatient data. The authors do not discuss what lay behind the improvement in HES data quality. Collection of sexual identity, gender, and behaviour, whilst lagging behind, have also been impacted by legislation that is incentivising data collection [33,62]. Furthermore, given the sensitive and private nature of information such as ethnicity, disability, gender, and sexual orientation, legal safeguards to ensure nondiscrimination on the basis of this information are also important factors that impact on data collection efforts [42,45].

### 3.2. Wider Actions to Enable Improvements in Data Quality

While mandating data collection leads to improvements in data quality, it needs to be supported by wider actions to enable organisations to put in place mechanisms to improve data quality at source [4,23,31,32]. Of the included reports, 38 provided evidence that achieving senior-level buy-in [4,34,42,45,65], the development of staff training programmes [19,20,22,24,25,27,29,31,32,35,36,37,49,54,58,61,62], guidance on how to use data [19,29,34,37], engagement activities with citizens, patients, and communities [17,25,29,49,56,58,65], and training on analysis of source data all contribute to efforts to improve data quality [19,20,24,25,27,29,31,32,35,36,54,58,61,62].

Senior-level buy-in is needed to prioritise data collection and put in place systems, such as IT infrastructure, to enable data collection, as well as utilisation of the data for service improvement. Davidson et al. (2021) report that obtaining executive-level buy-in was crucial for recording and improving ethnicity data collection in NHS Lothian [34]. Reports have shown that this can be achieved by demonstrating the value of data collection and analysis [19]. Using the data to demonstrate how outcomes or experiences vary for different groups, while also recognising the limitations of the data, created an awareness and interest in inequalities. This should result in an improvement spiral, driving a demand for better-quality data that in turn creates more interest in the intelligence based on that data [19,29]. Several papers reported the deliberations and recommendations of multidisciplinary groups created specifically to address issues in data quality in specific areas such as disability [38], paediatrics [57], deaf communities [18], and COVID-19 and ethnicity [65], or more broadly [68]. These examples demonstrate the value of multidisciplinary groups in informing efforts and developing effective solutions for improving data collection and analysis efforts.

Staff reluctance was cited in many reports as a key factor that may hinder attempts to improve data quality [4,19,20,29,71]. This was due to a lack of knowledge about the importance and use of the data, combined with staff reluctance to offend patients by asking for sensitive information. Training programmes were able to address this barrier and also assuage concerns relating to the use of systems to collect such data [20,22,24,25,27,29,31,35,49,54,58]. In addition, the development of guidance on using data was cited as a mechanism to improve data completeness and quality [4,34,43,68]. Training staff in communicating the rationale for data collection to the public and patients and on describing the parameters required was also a mechanism to improve data collection [34,60]. This was through building trust and openness between data collectors and providers [36,58]. One study suggested that ethnic matching could be one way of avoiding refusal during data collection [29].

In addition to staff reluctance, patients or the public may also be reluctant to provide data, or data collection instruments may not be appropriately developed for them. Several papers cited the importance of patient, public, or community involvement in initiatives to collect data or develop instruments such as surveys in data collection [27,58]. This involvement can help shape the questions that are asked and avoid marginalisation [17,36,38,56,63].

### 3.3. Data Collection Instruments, Systems and Standardisation

Many reports cited that data quality and granularity are impacted by the lack of standardised definitions. This creates pragmatic and logistical issues for data collection [19,21,71] through a lack of uniformity in data collection instruments such as surveys, as well as in IT systems that assign codes to different categories of data. Lack of standardised definitions and coding practices can cause major challenges when attempts are made to link data and in further analysis [63]. The introduction of standardised categories, or certain fields that are compulsory to complete as part of the design of IT systems, were mechanisms that were used to improve the recording and the quality of data [28,60,61,62]. Two papers recommended that consistency in coding and naming across different surveillance systems was also a way to enable consistency and more efficient linkage of sociodemographic data [23,25].

The importance of periodically revisiting these categories and ensuring their relevance was also shown to be an important activity [59]. Audit processes to monitor the completeness and accuracy of data and the methods used in data collection were discussed [20,70]. These processes allowed the assessment of data quality to put in place mechanisms for quality improvement [31]. One paper [16] reported on an instrument that could be used to compare and benchmark health information systems; however, it is unclear to what extent such tools are utilised or practical. Many grey literature reports in the UK recommended standardised protocols for collecting and recording ethnicity data as a mechanism to improve quality [4,65,67,70]. The importance of ensuring systems are in place to enable this was also discussed [31,38,63].

Improving the granularity and data fields available for individuals to self-assign their ethnicity or other characteristics was also shown to improve the completeness of data. For example, providing more options for self-reporting reduced the unknown ethnicity in certain studies [60]. This was achieved through providing more options (which are sometimes more relevant) to survey responders, resulting in less selection of the “unknown” category. Several reports used multidisciplinary groups to develop better understanding of the data that professionals from diverse disciplines thought should and can be collected [38,57,65,69].

### 3.4. Methodological Approaches to Improve Data Completeness and Accuracy

In addition to efforts to improve data at source, we also identified reports that described methods for improving data completeness and accuracy using statistical or other approaches (*n* = 27). This included data linkage, using proxy variables, or imputation through other methodologies [24].

Mathur et al. (2014) [47] found that when patients appeared in both the Clinical Practice Research Datalink (CPRD) and the Hospital Episode Statistics (HES) datasets with a usable ethnicity code in both datasets, the code was the same category in just 73% of cases. They found that when patients appeared in both datasets, completeness of usable ethnicity data in the CPRD increased from 78.7% to 97.1% once ethnicity data from HES was added. Knox et al. (2016) [45] looked at hospital admission rates by ethnicity in Scotland between 2009 and 2015, using the most recently recorded ethnicity to populate all admissions for that patient. This reduced the numbers of episodes with missing ethnicity from 24% to 15%, and the researchers completed the missing data for the remaining 15% by assigning those cases to ethnic groups in proportion to the distribution of known ethnicity by age and sex.

A number of imputation techniques can also be used to obtain more complete data; however, different methodologies have limitations and strengths [24,26]. Examples of the methods used include randomly assigning ethnicity, for example, on the basis of the distribution in the observed dataset or using a reference dataset [70], and using geographic location or probabilistic methods to infer ethnicity [35,50,51,58].

Several studies have investigated the use of algorithms to improve the completeness of ethnicity data by assigning ethnicity codes to individuals on the basis of their names, when self-identified data is missing [24,30,35,40,46,50,55,70]. The utility of this approach is recognised to differ considerably across countries because of significant variations in the composition of the population. Smith et al. (2017) [55] used the ‘Onomap’ software to categorise children and young people in the Yorkshire cancer registry as white, South Asian, or ‘other’ on the basis of their name, and also took ethnicity information from HES where this was recorded. Eleven per cent had missing HES ethnicity data and Onomap classified most of these patients. However, it is not clear whether these name-derived classifications were accurate, and these categories are very broad. The use of different methods to assign ethnicity did result in some different estimates of ethnic variation in cancer incidence, demonstrating the importance of accurate data.

Ryan et al. (2012) [50] also used Onomap and an additional name recognition software, Nam Pehchan [72], to predict the ethnicity of cases in a regional cancer registry who were missing this information following linkage with hospital inpatient data. They found that the software packages were accurate at predicting South Asian ethnicity but poor for other groups. They also looked at predicting ethnicity based on geographical area of residence but found this was also a poor predictor.

One paper also described the use of read codes to identify patients with learning difficulties (LD). NHS England issued guidance in October 2019 on improving the identification of people on the general practice LD register [69]. This required GPs to use a list of codes provided to check that all eligible patients were included on the practice LD register. The impact of this guidance on the numbers of patients on the register does not appear to have been evaluated. However, there was previous work evaluating the use of diagnostic read codes that found that this approach did identify small numbers of additional people who should have been on the register, and some further patients were found using specific descriptive codes [51]. The authors concluded that searching read codes to improve practice LD registers was quick and viable but not sufficient to capture most of the people eligible for inclusion, particularly those with milder learning difficulties. There does not appear to be evidence on how best to identify the remaining patients who could be included.

## 4. Discussion

Our scoping review identified a variety of mechanisms by which data quality in relation to health inequalities can be improved (Table 2). While the focus of many of the papers is on ethnicity data, many of the findings are also applicable to other dimensions of health inequalities because of the similarities in the issues that impact on data collection. There were relatively few papers that discussed improvements of data related to socio-economic status; however, this might be because such data are collected through other means, rather than self-reporting, and the practice for collating this data is better established. There were also relatively few papers that discussed improvement of data relating to gender and sexual orientation or disability. In addition, while some included papers discussed the issue of intersectionality, the impact in terms of data analysis or data collection were often not fully explored.

We have classified the mechanisms that can be used to improve the quality of data on health inequalities as more distal or proximal to the source data. Distal factors that impact on data quality include legislation and policies that are in place to ensure and mandate collection of data to enable addressing health inequalities. While many countries recommend the monitoring of data related to equality and discrimination, the extent to which this is implemented and actioned for health varies. Much of this is due to the differing structures of health systems and legislation that are in place globally. These distal factors impact on the ability to collect data related to equality and discrimination. For example, in the UK, the duty of data collection falls with public bodies [42], whereas this is not necessarily the case in other countries. Nevertheless, several included reports evidenced the fact that legalisation and policy were key contributors to the success of high-quality data collection efforts. Mechanisms to enact these policies and enable data collection form the next series of mechanisms to improve data quality. Reports described a variety of mechanisms, such as senior-level buy-in, staff training programmes, patient and public involvement, needed to enable creation of data systems that take into consideration the purpose of data collection and are timely and relevant.

Data pertaining to health inequalities may be collected by different organisations involved in health and care provision. They may collect these data for different purposes, meaning that the granularity of information requirements may differ. In addition, definitions in relation to many protected characteristics such as gender and ethnicity vary and evolve over time. This is because these are composite social constructs, attempting to bring together a number of different elements. For example, ethnicity is a composite of cultural factors, language, and ancestry, amongst others. This is evidenced by reports from the UK [4] that do not make a strong distinction between race and ethnicity, though work from the US distinguishes between these concepts, particularly when considering people from a Hispanic/Latinx background. Furthermore, these concepts change over time, meaning minority groups can change in size and new groups may become more prevalent. Many reports cited that redefinition of how populations are categorised in relation to characteristics related to health inequalities is needed over time [24,29,36]. For example, it is now more common to collect data that allows us to identify a subcategory of White Eastern European, or distinguish between Black African groups. Similarly, few would have included ‘nonbinary’ as a possible answer option to a question on gender five years ago. Thus, engagement across citizens, providers, and those creating data systems is needed to ensure the data that are collected are acceptable, relevant, and fit for purpose, and yet retain the ability to compare across time to monitor change and assess the impact of policies and interventions that aim to prevent and reduce health inequalities.

The report of improvements to data collected by NHS Lothian is a good example of the multi-layered approach that is needed to improve data quality [34]. The Scottish government and the Commission for Racial Equality requested the Scottish health boards to improve the recording of patient ethnicity data, and all boards were required to produce an action plan with progress measures. Davidson et al. (2020) report an impressive increase in the proportion of patients with a recording of ethnicity from 3% to over 90% in just three years (between 2008 and 2012). The authors attribute this improvement to several factors, chief amongst these being the decision to make ethnicity a mandatory field in the hospital data systems. Other important factors were thought to include the training of individuals responsible for data collection, awareness raising with relevant clinical and management staff and sharing a clear purpose and vision, and executive buy-in from senior clinical and management colleagues to ensure staff were able to prioritise recording these data. Making it clear to staff how ethnicity information is used was also important to maintain their motivation to collect these data. In this case, the data were used to demonstrate that rates of A&E use by ethnic minority groups did not appear to be linked to rates of registration in primary care. The progress made by NHS Lothian is in contrast to many other NHS Boards in Scotland where, over the same period, recording remained poor or improved much more slowly, despite an identical governance and legal context [73].

The importance of staff training is also evidenced by some older studies. A review by Iqbal et al. (2009) showed that staff training was the main intervention for which there was evidence of data quality improvements for patient ethnicity, followed by adequate resources to allow data collection and use [74]. Training should be tailored to the local context and explain why it is important to gather standardised data on patient ethnicity, what the data will be used for, and how to ask the questions and record responses. The review also recommended collecting self-reported ethnicity as routine during GP registration.

Self-reported data are the gold standard for certain data such as gender and ethnicity that can inform studies of health inequality. However, the work included for this review has identified a wide range of reasons why individuals may be reluctant to share personal data relevant to these characteristics. A paper from NHS Scotland points out that different settings can have substantially different rates of refusal (for ethnicity data reporting), which suggests different organisational approaches to asking for and recording the information [70]. High rates of refusal (or high use of an ‘other’ category) can be compared against peer organisations and could likely be brought down by learning from successful approaches elsewhere. Improving public and patient understanding of why this information is being collected and how it will be used can also encourage efforts to improve data collection and, therefore, quality. Nevertheless, there will likely always be some people who decline to give information on their ethnicity, or other personal information not perceived to be directly relevant to their immediate care, and it is important to recognise their right to decline to provide this.

It can take time to put in place a best practice that leads to the collection of good-quality data in relation to health inequalities. In addition, as evidenced by many of the reports, this may still lead to incomplete data with inaccuracies. Thus, mechanisms that can improve the accuracy, quality, and completeness of available data are also important. We identified studies that reported the use of methodologies such as linkage, imputation, and the use of proxy variables. However, there are several limitations to these methods. Using naming software or linked data to improve the completeness of ethnicity data takes considerable time and analytical expertise and is not ideal for producing useful up-to-date routine reports for health services [50,55]. However, the studies that examined the use of naming software took place at a time when recording ethnicity for hospital inpatients was much poorer. It seems likely that their findings have less relevance today when hospital data are much less likely to be missing data on patient ethnicity, given that both studies were of cancer patients (who are likely to appear in HES data). Using naming software to estimate ethnicity may still have utility when data cannot be linked to hospital or other data, but clearly this approach to filling ethnicity-data gaps needs caution. It is likely to struggle more with mixed-ethnicity individuals (an increasing proportion of the UK population) and is unlikely to be able to produce the detail necessary to distinguish between subgroups.

Data linkage has been evaluated for its utility in reducing missing data. If the same individual is identifiable in two datasets, information from one dataset can be used to check or complete the information in the other. Data linkage can be powerful for ‘filling in the gaps’ and has been used by NHS Digital to increase coverage of ethnicity data during the COVID-19 pandemic [75]. However, using data linkage to improve ethnicity data on a routine basis, so that it can be useful for producing near-real-time intelligence to inform services and policy, is challenging given the requirements for analyst capacity and time [24]. Improving data through data linkage also requires having a resource to link to that contains accurate self-reported ethnicity data and has high coverage across the population. In England, this resource could potentially be census data, HES data, or GP data, or death certification data for people who have died. However, there are issues with each of these sources. Census data is very sensitive and not easy to access and is only updated every 10 years. GP data is known to have patchy coverage. Recent HES data has better completeness for the people included in the dataset, but coverage is an issue because of the requirement that patients have been hospital users. Using ethnicity data from death certificates is also likely to bring accuracy issues as, of course, ethnicity cannot be self-reported in these cases and, in fact, often mismatches the data in hospital records. Even within the group of patients who appear in the HES data, using HES as a source of accurate ethnicity data may be inadequate.

This scoping review has some strengths in that we used a systematic approach to identify as many reports as possible discussing different mechanisms to improve data quality. Yet, it is likely that there are reports that we missed, especially in the form of grey literature, because of the broad nature of the subject matter. The majority of the reports were from the UK or US. This might be a result of our search terms not being optimal. Other factors include the extent to which health inequalities monitoring has been implemented and is a priority as part of healthcare delivery [76,77]. Nevertheless, this work identified evidence for several distal and proximal approaches that can be taken within the healthcare context of the United Kingdom to improve the quality of the data used for the identification and monitoring of health inequalities. Some of these approaches may be transferable to other healthcare contexts. However, given differences in definitions and drivers of health inequalities and provision of health care around the globe, they may not apply to the same extent.

## 5. Conclusions

Accurate and timely data are essential in identifying inequalities in health and care, in understanding where inequalities occur and which groups are affected, and in assessing the impact of interventions. Despite this, many health-related datasets either do not routinely collect important dimensions of inequality or are limited by poor-quality data. Where data are available, they may not always be used to the best extent. Our review identified that a variety of effective mechanisms are available and can be utilised to improve data quality. These include those that are distal and impact on data collection, or those that are more proximal to the source data and can aid in data analysis. Given the renewed emphasis on the need for action to address health inequalities at both a national and a system level, it is important to understand how systems can easily implement the mechanism described in our review. This will likely require working with senior leaders, staff, and analysts to gain buy-in and identify effective ways to implement mechanisms to address issues with data quality. Further work is underway to understand how best to support health and care staff to act on the evidence identified in this review to improve the quality of data relating to health inequalities within their organisations and local systems.

## Figures and Tables

**Figure 1 ijerph-19-15874-f001:**
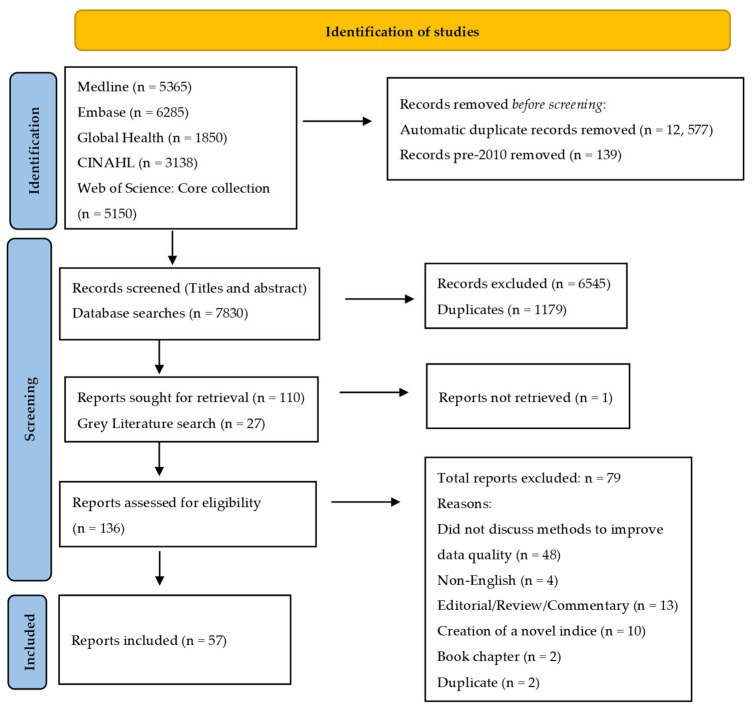
Flow diagram of included and excluded studies.

**Table 1 ijerph-19-15874-t001:** Included studies and characteristics.

Author, Year	Title	Type of Data Discussed	Distal Factors	Wider Actions to Enable Improvements in Data Collection	Data Collection Instruments, Systems, and Standardisation	Methodological Approaches to Improve Data Quality and Accuracy
Abouzeid, M et al. 2014 [16]	The potential for measuring ethnicity and health in a multicultural milieu—the case of type 2 diabetes in Australia	Ethnicity	✓	✓	✓	✓
Allen, VC et al. 2011 [17]	Issues in the Assessment of “Race” Among Latinos: Implications for Research and Policy	Ethnicity	✓	✓	✓	
Anderson, ML et al. 2018 [18]	Deaf Qualitative Health Research: Leveraging Technology to Conduct Linguistically and Sociopolitically Appropriate Methods of Inquiry	Disability		✓	✓	
Andrews, RM 2011 [19]	Race and Ethnicity Reporting in Statewide Hospital Data: Progress and Future Challenges in a Key Resource for Local and State Monitoring of Health Disparities	Ethnicity	✓	✓	✓	
Azar, KMJ et al. 2012 [20]	Accuracy of Data Entry of Patient Race/Ethnicity/Ancestry and Preferred Spoken Language in an Ambulatory Care Setting	Ethnicity		✓	✓	
Becker, T et al. 2021 [21]	Data Disaggregation with American Indian/Alaska Native Population Data	Ethnicity	✓	✓	✓	✓
Berry C et al. 2013 [22]	Moving to patient reported collection of race and ethnicity data Implementation and impact in ten hospitals	Ethnicity			✓	
Beltran VM et al. 2011 [23]	Collection of social determinants of health measures in U.S. national surveillance systems for HIV, viral hepatitis, STDs, and TB	Infectious disease	✓	✓	✓	✓
Bilheimer LT et al. 2010 [24]	Data and Measurement Issues in the Analysis of Health Disparities	General	✓	✓	✓	✓
Block RG et al. 2020 [25]	Recommendations for improving national clinical datasets for health equity research	General		✓	✓	✓
Blosnich JR et al. 2018 [26]	Using clinician text notes in electronic medical record data to validate transgender-related diagnosis codes	Gender				✓
Bozorgmehr, K et al. 2017 [16]	How Do Countries’ Health Information Systems Perform in Assessing Asylum Seekers’ Health Situation? Developing a Health Information Assessment Tool on Asylum Seekers (HIATUS) and Piloting It in Two European Countries	General			✓	
Cahill SR et al. 2016 [27]	Inclusion of Sexual Orientation and Gender Identity in Stage 3 Meaningful Use Guidelines: A Huge Step Forward for LGBT Health	Gender	✓			
Chakkalakal RJ et al. 2015 [28]	Standardized Data Collection Practices and the Racial/Ethnic Distribution of Hospitalized Patients	Ethnicity			✓	✓
Chen Y et al. 2018 [29]	Racial Differences in Data Quality and Completeness: Spinal Cord Injury Model Systems’ Experiences	Ethnicity		✓	✓	
Clarke LC et al. 2016 [30]	Validity of Race, Ethnicity, and National Origin in Population-based Cancer Registries and Rapid Case Ascertainment Enhanced with a Spanish Surname List	Ethnicity				✓
Craddock L et al., 2016 [31]	Assessing race and ethnicity data quality across cancer registries and EMRs in two hospitals	Ethnicity	✓		✓	
Cruz, TM 2020 [32]	Perils of data-driven equity: Safety-net care and big data’s elusive grasp on health inequality	General		✓		
Cruz, TM 2021 [33]	Shifting Analytics within US Biomedicine: From Patient Data to the Institutional Conditions of Health Care Inequalities	General	✓	✓	✓	
Davidson E et al., 2021 [34]	Raising ethnicity recording in NHS Lothian from 3% to 90% in 3 years: processes and analysis of data from Accidents and Emergencies	Ethnicity	✓	✓		✓
Derose, SF et al. 2013 [35]	Race and Ethnicity Data Quality and Imputation Using US Census Data in an Integrated Health System: The Kaiser Permanente Southern California Experience	Ethnicity			✓	✓
Donald C and Ehrenfeld JM 2015 [36]	The Opportunity for Medical Systems to Reduce Health Disparities Among Lesbian, Gay, Bisexual, Transgender and Intersex Patients	Gender	✓	✓	✓	
Escarce, J et al. 2011 [37]	Collection Of Race and Ethnicity Data By Health Plans Has Grown Substantially, But Opportunities Remain To Expand Efforts	Ethnicity	✓	✓	✓	
Fortune, N et al. 2020 [38]	The Disability and Wellbeing Monitoring Framework: data, data gaps, and policy implications	Disability	✓	✓	✓	✓
Frank, J and Haw S 2011 [39]	Best Practice Guidelines for Monitoring Socioeconomic Inequalities in Health Status: Lessons from Scotland	Gender		✓	✓	✓
Fremont, A et al. 2016 [40]	When Race/Ethnicity Data Are Lacking: Using Advanced Indirect Estimation Methods to Measure Disparities	Ethnicity				✓
Haas, AP et al. 2015 [41]	Collecting Sexual Orientation and Gender Identity Data in Suicide and Other Violent Deaths: A Step Towards Identifying and Addressing LGBT Mortality Disparities	Gender			✓	
Hannigan, A et al. 2019 [42]	Ethnicity recording in health and social care data collections in Ireland: where and how is it measured and what is it used for?	Ethnicity	✓	✓		✓
Jorgensen S et al., 2010 [43]	Responses of Massachusetts hospitals to a state mandate to collect race, ethnicity and language data from patients: a qualitative study	Ethnicity	✓	✓	✓	
Khunti, K et al. 2021 [44]	The need for improved collection and coding of ethnicity in health research	Ethnicity			✓	
Knox et al. 2019 [45]	The challenge of using routinely collected data to compare hospital admission rates by ethnic group: a demonstration project in Scotland	Ethnicity	✓			✓
Liu, L et al. 2011 [46]	Challenges in Identifying Native Hawaiians and Pacific Islanders in Population-Based Cancer Registries in the U.S.	Ethnicity				✓
Mathur et al., 2013 [47]	Completeness and usability of ethnicity data in UK-based primary care and hospital databases	Ethnicity	✓			✓
Pinto, AD et al. 2016 [48]	Building a Foundation to Reduce Health Inequities: Routine Collection of Sociodemographic Data in Primary Care	General		✓	✓	
Polubriaginof, FCG et al. 2019 [49]	Challenges with quality of race and ethnicity data in observational databases	Ethnicity		✓	✓	
Ryan et al. 2012 [50]	Use of name recognition software, census data and multiple imputation to predict missing data on ethnicity: application to cancer registry records	Ethnicity				✓
Russell AM et al. 2017 [51]	Identifying people with a learning disability: an advanced search for general practice	Learning disability				✓
Saperstein, A. 2012 [52]	Capturing complexity in the United States: which aspects of race matter and when?	Ethnicity			✓	
Shah, SN et al. 2014 [53]	Measuring and Monitoring Progress Toward Health Equity: Local Challenges for Public Health	General	✓	✓	✓	✓
Siegel, B et al. 2012 [54]	A Quality Improvement Framework for Equity in Cardiovascular Care: Results of a National Collaborative	General		✓	✓	
Smith L et al., 2017 [55]	Comparison of ethnic group classification using naming analysis and routinely collected data: application to cancer incidence trends in children and young people	Ethnicity				✓
Smylie, J and Firestone M 2015 [56]	Back to the basics: Identifying and addressing underlying challenges in achieving high quality and relevant health statistics for indigenous populations in Canada	Ethnicity		✓		
Tan-McGrory, A et al. 2018 [57]	A patient and family data domain collection framework for identifying disparities in pediatrics: Results from the pediatric health equity collaborative	General		✓	✓	
Thorlby R et al. 2011 [58]	How Health Care Organizations Are Using Data on Patients’ Race and Ethnicity to Improve Quality of Care	Ethnicity	✓	✓	✓	✓
Wang KR et al. 2020 [59]	Information Loss in Harmonizing Granular Race and Ethnicity Data: Descriptive Study of Standards	Ethnicity		✓	✓	
Webster P and Sampangi S, 2014 [60]	Did We Have an Impact? Changes in Racial and Ethnic Composition of Patient Populations Following Implementation of a Pilot Program	Ethnicity	✓	✓	✓	
Wei-Chen, L et al. 2016 [61]	Improving the Collection of Race, Ethnicity, and Language Data to Reduce Healthcare Disparities: A Case Study from an Academic Medical Center	General		✓	✓	
Wolff, M et al. 2017 [62]	Measuring Sexual Orientation: A Review and Critique of US Data Collection Efforts and Implications for Health Policy	Gender	✓	✓	✓	✓
Zhang XZ et al. 2019 [63]	Role of Health Information Technology in Addressing Health Disparities Patient, Clinician, and System Perspectives	General		✓	✓	✓
Hutt P and Gilmour S 2010 [64]	Tackling inequalities in general practice	General		✓	✓	
Scottish Government 2020 [65]	Improving data and evidence on ethnic inequalities in health: Initial advice and recommendations from the expert reference group on ethnicity and COVID-19	Ethnicity	✓	✓	✓	✓
NHS England, 2020 [66]	Advancing mental health equalities strategy	General	✓		✓	
NHS, 2019 [67]	NHS Mental Health Implementation Plan 2019/20–2023/24	General	✓		✓	
NHS Race and Health Observatory, 2021 [68]	Ethnic health inequalities and the NHS: Driving progress in a changing system	Ethnicity	✓	✓	✓	
Scobie S, Spencer J and Raleigh V, 2021 [4]	Ethnicity coding in English health service datasets	Ethnicity	✓	✓	✓	✓
NHS England, 2019 [69]	Improving identification of people with a learning disability: guidance for general practice	Learning disability		✓	✓	
National Services Scotland, 2017 [70]	Measuring use of health services by equality group	General		✓	✓	✓
Total citations		57	26	28	43	27

**Table 2 ijerph-19-15874-t002:** Summary of best practices.

Theme	Point in the Data Pathway	Actions
Distal factors	Upstream of data collection and analysis	Mandating data collection
		Legal safeguards to ensure nondiscrimination
		Legislation incentivising data collection
		Prioritisation in policy
Wider actions to enable improvements in data collection	Preparing for data collection	Achieving senior-level buy-in in organisations involved in data collection
		Engagement activities with citizens, patients, and communities
		Staff training programmes on purpose and mechanisms for data collection
		Developing guidance on how data can be used
		Demonstration of the value of data collection and analysis for organisations
Data collection instruments, systems, and standardisation	Data collection	Using multidisciplinary groups to inform data collection instruments, systems, and standardisation
		Creating standardised definitions and coding practices across organisations
		Improving granularity of data fields
		Developing standardised processes for collecting and recording data
		Developing audit processes to monitor data quality aspects
		Creating IT systems to facilitate data collection
		Periodic revision of definitions and categories
Methodological approaches to improve data quality and accuracy	Data analysis	Linking with other data sources
		Use of proxy variables
		Imputation

## Data Availability

Data can be made available on request.

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
