# Peer review of "A Scoping Review of Approaches to Improving Quality of Data Relating to Health Inequalities"

_ijerph, 2022, doi:10.3390/ijerph192315874_

Round 1

Reviewer 1 Report

The purpose of this paper is to identify strategies to improve quality and specificity of health related data by reviewing the literature for effective practices. This article is comprehensive and generally well-written. I have only 2 comments.

1. The background can be reduced by a paragraph if the authors use more directive statements, less 'lead in' sentences. For example, Line 35 reduce to "lead healthy lives, including differences....

Line 44: Take out 'it is recognized' - what with However, the quality...

lines 47-60: Lots of examples here - state what you want to state. Reduce the number of words. It will make the case much stronger.  Same throughout the background in particular.

2. I think an itemized list or maybe a small table with the 'best practices' would be very useful to readers.

Reviewer 2 Report

Thank you for providing an opportunity to review the manuscript. This is an interesting study that provides a systematic scoping review of approaches to improving quality of data relating to health inequalities. The text is relatively well written; however, it needs some minor improvement:

1)    The introduction is incomplete. Authors should provide the definition of “quality of data”

2)    Authors found that most articles were from high income countries (US and UK) where the health inequalities are lower than low and middle income countries. It would be great if the authors can add the articles from low- and middle-income countries or discussion and explain more in the limitation.

3)    Authors should mention on specific areas and articles for example, infectious diseases, learning disabilities or cardiovascular care in the results and discussion.

4)    In the limitation, please provide in more detailed such as the result can not represent in whole population but specifically in high income countries, it may not be representative of all available literature in the field.

5)    Authors may check the presentation of the results followed the checklist for reporting of a scoping review from PRISMA guideline or explain how authors present the report.

Reviewer 3 Report

Exploring health inequalities and their causes is an important addition to the understanding of social and income differences, and its public health significance cannot be denied in times of crises, when access to health and care is limited. The authors' meta-analysis focuses on the quality and reliability of the data and data collection on the subject and correctly lists the common problems and recommends appropriate mechanisms to eliminate them. It also affects sensitive patient rights issues.

This paper is a relevant methodological guide for researchers working in the field, so I fully support its publication. One question remained open, why were only studies completed before 2010 evaluated and excluded those published afterwards? Health inequalities have increased significantly in the last 5 years and the recently conducted investigations can also contribute to crisis management strategies.

Reviewer 4 Report

Thank you for this coherent and comprehensive review.

The use of any guidelines like the PRISMA-SR are not reported e.g.  Tricco, AC, Lillie, E, Zarin, W, O'Brien, KK, Colquhoun, H, Levac, D, Moher, D, Peters, MD, Horsley, T, Weeks, L, Hempel, S et al. PRISMA extension for scoping reviews (PRISMA-ScR): checklist and explanation. Ann Intern Med. 2018,169(7):467-473. doi:10.7326/M18-0850.

I am not sure if the Journal requires this- but many Journals do these days and is suspect many for the criteria have been met? I wonder if i a statement about this might be worthwhile.

There was no critique of the included papers. Again, this is not always completed with a scoping review. but it would be good to include the rationale for not doing so.

There was no critical discussion about the frequency of types of data collected in the manuscript. I think this is an important discussion point as it perhaps reflects what aspects of 'ism' we as a society are more comfortable to start exploring and monitoring. It is also important because of the impact of intersectionality. If some aspects of identity are not collected (e.g. disability or gender) then the true effect of health inequities are not observed. Could the authors please make some comments on this query and some recommendations going forward.

Could more information about what was covered in the 'general' category please be provided. 

Thank you for the paper.
